# Dissection of Metabolome and Transcriptome—Insights into Capsaicin and Flavonoid Accumulation in Two Typical Yunnan Xiaomila Fruits

**DOI:** 10.3390/ijms25147761

**Published:** 2024-07-16

**Authors:** Huaran Hu, Lei Du, Ruihao Zhang, Qiuyue Zhong, Fawan Liu, Weifen Li, Min Gui

**Affiliations:** Horticultural Research Institute, Yunnan Academy of Agricultural Science, Kunming 650205, China; hhuaran@outlook.com (H.H.); sxdl8882024@163.com (L.D.); zrh@yaas.org.cn (R.Z.); yyskgk@yaas.org.cn (Q.Z.); 13888826387@163.com (F.L.); 15825259917@163.com (W.L.)

**Keywords:** pepper, capsaicin, flavonoid, phenylpropanoid, metabolome, transcriptome, transcription factor

## Abstract

Pepper is an economically important vegetable worldwide, containing various specialized metabolites crucial for its development and flavor. Capsaicinoids, especially, are genus-specialized metabolites that confer a spicy flavor to *Capsicum* fruits. In this work, two pepper cultivars, YB (*Capsicum frutescens* L.) and JC (*Capsicum baccatum* L.) pepper, showed distinct differences in the accumulation of capsaicin and flavonoid. However, the molecular mechanism underlying them was still unclear. Metabolome analysis showed that the JC pepper induced a more abundant accumulation of metabolites associated with alkaloids, flavonoids, and capsaicinoids in the red ripening stages, leading to a spicier flavor in the JC pepper. Transcriptome analysis confirmed that the increased expression of transcripts associated with phenylpropanoid and flavonoid metabolic pathways occurred in the JC pepper. Integrative analysis of metabolome and transcriptome suggested that four structural genes, *4CL7*, *4CL6*, *CHS*, and *COMT*, were responsible for the higher accumulation of metabolites relevant to capsaicin and flavonoids. Through weighted gene co-expression network analyses, modules related to flavonoid biosynthesis and potential regulators for candidate genes were identified. The promoter analysis of four candidate genes showed they contained several cis-elements that were bonded to MYB, bZIP, and WRKY transcription factors. Further RT-qPCR examination verified three transcription factors, MYB, bZIP53, and WRKY25, that exhibited increased expression in the red ripening stage of the JC pepper compared to YB, which potentially regulated their expression. Altogether, our findings provide comprehensive understanding and valuable information for pepper breeding programs in the future.

## 1. Introduction

Chili pepper (*Capsicum* spp.) is one of the ancient vegetables consumed in humans’ daily diet [1]. With increasingly frequent economic and cultural exchanges, peppers spread around the world and evolved into new breeds, contributing to their adaption to different agroclimatic regions and directional selection in different areas such as food, medicine, and ornamentals [2]. Hitherto, more than 32 species have been identified in the Capsicum breeds, and five genera of peppers, *C. annuum* L., *C. baccatum* L., *C. chinense* Jacq., *C. frutescens* L., and *C. pubescens* (Ruiz and Pavon), were determined to be the most planted cultivars in the world after normalization [3]. It has been proven that pepper cultivars and ripening stages affect the accumulation of flavors and metabolites [4,5].

As an important economic crop, peppers possess a special flavor and abundant nutrition, especially satisfying people’s spicy taste [6]. The pungency of pepper is due to the accumulation of capsaicinoids (CAPDs) belonging to a group of alkaloids that are unique to the Capsicum genus. Capsaicin (CAP), dihydrocapsaicin (DhCAP), and nordihydrocapsaicin (NDhCAP) constitute the primary capsaicinoids, which are exclusively produced in the pepper fruit. Increasing evidence confirmed that these metabolites are produced through a complex series of reactions involving portions of the flavonoid and phenylpropanoid metabolic pathways [7,8,9]. With advanced molecular biology in recent years, many researchers have been devoted to identifying and deciphering the essential structural genes involved in capsaicin biosynthesis. *Phenylalanine ammonia-lyase (PAL)*, *cinnamate 4-hydroxylase (C4H)*, *4-coumaroyl-CoA ligase (4CL)*, and *caffeoyl-CoA 3-O-methyltransferase (COMT)* have been universally documented and verified in the regulation of certain secondary metabolites, which acted in important roles in pathogen resistance, abiotic stress and flavor formation, and flower coloration in multiple plants [10,11,12]. Additionally, the genes including *branched-chain amino acid aminotransferase (BCAT)*, *acyl-ACP thioesterase (FatA)*, and *3-ketoacyl-ACP synthase (KAS)* and *Capsaicinoid synthase (CS*) were also identified in further research [13,14]. These necessary genes for capsaicinoid biosynthesis significantly altered the pungent degree in different pepper cultivars depending on their transcriptional abundance.

The capsaicinoid biosynthesis pathway showed the convergency of phenylpropanoid and flavonoid pathways. Thus, the accumulated metabolites shared in the metabolic pathways could affect multiple physiological traits in Solanaceae plants. And soluble sugars, organic acids, phenolics, and volatiles are important components that determine unique fruit flavors and consumer preferences. Thus, these flavor traits were also taken into consideration in the pepper fruit besides spiciness. Studies have shown that the abundant accumulation of flavanone compounds in the *Citrus genus* determined their fruit flavor and were beneficial for human health [15]. The accumulation of flavonoids (flavanol and anthocyanin) and phenolics (epicatechin, catechin, kaempferol, chlorogenic acid, and caffeic acid) greatly affected the fruit flavor in apple and grapefruits [16,17]. Moreover, flavonoid and phenylpropanoid metabolic pathways were well implicated in the confrontation of adverse stress, including low temperature, ultraviolet, drought, and pathogen invasion [18,19]. These studies stressed the importance of flavonoid and phenylpropanoid biosynthesis in breeding improvement and highlighted their value in further molecular breeding programs.

With the development of modern molecular biology, the integration of multi-omics analyses, such as metabolomics and transcriptomics, has been proven to be an effective approach to decipher the flavor metabolic pathways and identify regulatory genes [20,21]. MYB, ERF, as well as bHLH, bZIP and MADS-box, have been reported to regulate the gene expression involved in capsaicinoid and flavonoid biosynthesis [22,23,24,25,26]. The application is critical for deciphering the flavor-associated metabolome and transcriptome patterns in the pepper fruit.

Xiaomila (*C. frutescens* L.) is a unique germplasm resource of pepper in China; Yunnan province is its predominant cultivation region, with the largest production and processing center of Xiaomila in China. Xiaomila has gained popularity for its unique spicy flavor and abundant nutrients. It was commonly used as pickled pepper (Paojiao) to produce appetizers and condiments, which has become an important source of the core flavor of Yunnan cuisine [27]. YB, as a local conventional variety, has been widely planted in Yunnan province. However, there are some problems in the process of industrialization, such as variety degradation, low yield, hysteretic growth period, and weakened resistance. Importantly, it was difficult to perform artificial pollination and pollen collection due to its small flower. Therefore, the limited agronomic measures and breeding techniques for YB Xiaomila merely maintained its purification and sub-generation. To address this issue, a new variety of Xiaomila designated as JC (Jingcui) based on the hybridization of *C. baccatum* × *C. baccatum* was generated. The JC pepper exerted excellent performance, such as increased fruiting rate, elevated resistance, shorter developmental period, and special fruit flavor, compared to the traditional local YB variety. In addition, because of their thick flesh, JC is more suitable for the raw material of pickled peppers. As a new type of Xiaomila, JC has now been widely promoted and cultivated in Yunnan. However, the differences in functional components and the molecular mechanism underlying these ameliorative traits remained poorly understood. There is a lack of research into multi-omics approaches to studying the flavor of Xiaomila.

In this study, our work aimed to compare the differences in flavor accumulation between two pepper genotypes at different development stages by multi-omics analysis. Herein, we developed the integrated analysis of transcriptome and metabolome profiles for YB and JC to reveal the metabolic changes and to identify regulatory genes associated with flavor substances. This study expects to provide valuable insights into the molecular breeding of peppers, targeting the improvement of pepper flavor.

## 2. Results

### 2.1. The Determination of Main Flavor Compounds of YB and JC in the Green and Red Mature Stages

To assess the flavor value of both Xiaomila cultivars, the green and red mature stages of both pepper cultivars were extracted for flavor detection (Figure 1A). In detail, the flavor-associated substances, including flavonoids, phenols, vitamin C, nor-dihydro-capsaicin, and dihydro-capsaicin, were fully measured. The JC pepper contained a more abundant accumulation of flavonoids, phenols, vitamin C, and capsaicin content (such as nor-dihydro-capsaicin and dihydro-capsaicin) compared to the YB pepper (Figure 1B). Especially in the red mature stage, the spike was more conspicuously observed in the accumulation of capsaicin and total phenols than the other index in the JC pepper. Intriguingly, we noticed that the spicy flavor-related substance, including capsaicin, nor-dihydro-capsaicin, and dihydro-capsaicin, was significantly increased in the JC pepper, demonstrating that the JC pepper exhibited a more pungent flavor than the YB pepper. Overall, the increased accumulation of the above compounds conferred excellent flavor in the JC pepper.

### 2.2. The Global Metabolome Analysis of Both Cultivars

To further investigate the potential mechanism of the accumulated flavor substance in the JC pepper, a systematical metabolomic analysis was performed. Statistically, a total of 1028 members of metabolites were identified in secondary-level metabolite analysis. After filtration with the elimination of unrecognized metabolites, 869 metabolites were obtained in the final, including hydrocarbons, benzenoids, lipids, organic acids, and alkaloids (Figure 2A and Appendix A). PCA plots exhibited the separated distribution of metabolic profiles in different mature stages of both pepper cultivars, suggesting credible analysis in follow-up determination (Figure 2B,C). Consistently, the unsupervised correlation analysis results showed marked aggregations and separations among four groups, in agreement with observations from the PCA diagrams (Appendix A). Moreover, PLS-DA validation authorized significant metabolite changes between the pairwise comparison of groups in both pepper cultivars, suggesting VIP analysis could be credible to screen differentially expressed metabolites (DEMs) (Appendix A).

Further, the volcano plots depicted the DEMs with the strict screening standard of q-value < 0.05 in pairwise comparisons of both pepper cultivars (Figure 2D). The results showed that 283 up-regulated DEMs and 298 down-regulated DEMs were identified in YB-green vs. YB-red. We noticed that the main changeable components came from organic acids, glycerophospholipids, fatty acyls, and flavonoids (Figure 2E), which was consistent with metabolite constitution during JC development (Figure 2G,H). These metabolites partook in the regulation of fruit flavor, coloration, and quality, which may explain the major flavor differences between two peppers.

Subsequently, we further investigated the functional distribution of DEMs, showing that DEMs relevant to amino acid metabolism, phenylpropanoid biosynthesis, flavonoid biosynthesis, and glycerophospholipid metabolism were significantly enriched in the comparison of YB-green vs. YB-red (Figure 2F,I). Consistently, we found similar enrichment terms of metabolic pathways in JC-green vs. JC-red. This convergent evidence suggested that active energy metabolism (purine, glycerophospholipid, pyrimidine) and secondary metabolism (phenylpropanoid and flavonoid biosynthesis) mainly promoted the accumulation of flavor substance during red mature stage in both pepper cultivars. Thus, our further work focused on the changes in metabolic pathways, including phenylpropanoid and flavonoid biosynthesis, which caused the formation of flavor substances in both pepper cultivars.

### 2.3. JC Induced Higher Accumulation of Metabolites Related to Flavonoids and CAPs in Both Ripening Stages

Given the marked advantage of the JC pepper that the higher content of the flavor-associated substance in the red mature stage, we aimed to investigate the differences of metabolome patterns in the green and red mature stages of both pepper cultivars by pairwise comparisons. We conducted the PCA analysis to assess the differences in metabolic profiles between both peppers, showing diverse metabolite components between both peppers during their growth development (Figure 3A,B). A total of 273 up-regulated DEMs and 269 down-regulated DEMs were identified in YB-green vs. JC-green (Figure 3C), and we found that metabolites, including benzenoids, organic acids, glycerophospholipids, and flavonoids, represented the main changes in comparison due to their large occupation of all DEMs (Figure 3D). We consistently acquired similar metabolite constitutions in YB-red vs JC-red (Figure 3E,F). In common, this evidence confirmed that increased accumulation of organic acids and flavonoids may cause the preferred flavor in the JC pepper.

Then, the DEMs were imported into KEGG enrichment analysis, showing that DEMs related to glycerophospholipid metabolism, amino acid metabolism, phenylalanin biosynthesis, and alkaloid biosynthesis, citrate cycle (TCA cycle) were significantly enriched (*p*-value < 0.05) in YB-green vs. JC-green (Figure 3G). Likewise, DEMs were remarkably gathered (*p*-value < 0.05) in metabolic pathways, including glycerophospholipid metabolism, phenylpropanoid biosynthesis, amino acid metabolism, and starch and sucrose metabolism in the comparison of YB-red vs. JC-red (Figure 3H).

In common, we found that most of the DEMs were mainly enriched in carbon source metabolism and amino acid metabolism in both pairwise comparisons (45/49 metabolites in YB-green vs. JC-green and 29/37 metabolites in YB-red vs. JC-red) and functioned with more active form in the JC pepper (Appendix A). These active metabolic pathways contributed to providing abundant substrates for the biosynthesis of flavor substances. Additionally, we noticed that two flavor-related pathways, flavone and flavonol biosynthesis and phenylpropanoid biosynthesis, were significantly enriched in the JC pepper. And a much larger number of DEMs were enriched in the red mature stage compared with the green mature stage in pairwise comparisons. In agreement with previous results (Figure 2), the results confirmed that the JC pepper possesses the advantages of highly accumulated flavor substance, which is attributed to more active energy, phenylpropanoid biosynthesis, and flavonoid biosynthesis metabolic pathways.

### 2.4. The Abundant Accumulation of CAPs and Alkaloids Facilitated the Spicy Flavor in JC Pepper

To further explain the advantage of abundant accumulation of flavor substance in the JC pepper, we decided to compare the up-regulated DEMs in pairwise comparisons (Figure 3) and plotted their distributions in Venn diagrams. A total of 78 up-regulated DEMs were shared in the comparison of YB-red vs. JC-red and YB-green vs. JC-green (Figure 4A). They were mapped into KEGG enrichment analysis, illustrating that these DEMs were significantly enriched (*p*-value < 0.05) in amino acid metabolism, glycerophospholipid metabolism, phenylalanine metabolism, alkaloid metabolism, and flavonoid biosynthesis (Figure 4B).

In particular, we noticed that DEMs related to flavonoid biosynthesis, phenylalanine metabolism, and alkaloid biosynthesis hold the major occupation (Appendix A). A heatmap depicting their expression among four groups showed that a total of 35 metabolites were identified, and they were divided into three types, alkaloids, flavonoids, and capsaicin (Figure 4C). Obviously, spicy flavor-related metabolites, including CAP, capsinate, DhCAP, dinorcapsaicin, homocapsaicin, norcapsaicin, and NDhCAP, were markedly up-regulated in the JC-red group. As a capsaicinoid-like compound, capsinate shares a similar biological structure with capsaicinoids but shows low pungency. This evidence confirmed that the JC pepper exerted a spicier flavor than the YB pepper due to the high accumulation of capsaicin in the red mature stage. Meanwhile, we noticed an increased accumulation of amino acids and their derived metabolites in JC-red samples, including isoleucine, aspartic acid, 6-hydroxynicotinic acid, aspartate, 4-hydroxybenzoic acid, and N-acetyl-L-phenylalanine. The accumulation of these metabolites may facilitate the biosynthesis of flavor substances as substrates or functions in other biological processes in the JC pepper.

### 2.5. Different Transcriptome Patterns Caused the Variable Accumulation of Flavor Substance between Both Cultivars

To further explore the differences in the regulation mechanism of flavor substance in both pepper cultivars, RNA-seq analysis was performed. We first assessed the differences in transcriptome patterns among four groups by PCA analysis, showing the disparate distribution between pairwise comparisons in both pepper cultivars (Figure 5A). Further Pearson correlation analysis verified that remarkable aggregations were observed in both pepper cultivars of four groups, which supported the PCA results (Figure 5B). The result confirmed that the changed transcriptome patterns were mainly responsible for the accumulation of flavor substances during the mature transformation of both pepper cultivars. Consistent with previous metabolome analysis, these results showed that the red mature stage was the essential period, resulting in the accumulation of flavor substances in both pepper cultivars.

Subsequently, volcano plots showed that a total of 3408 up-regulated DEGs and 2470 down-regulated DEGs were identified in YB-green vs. JC-green, as well as 4835 up-regulated DEGs and 4023 down-regulated DEGs in YB-red vs. JC-red (Figure 5C). We then linked these DEGs to the KEGG enrichment pathway, showing that the DEGs were significantly enriched in the photosynthesis, mismatch repair, DNA replication, nucleotide excision repair, plant-pathogen interaction, and homologous recombination metabolic pathways in the comparison of YB-green vs. JC-green (Figure 5D). In parallel, the DEGs involved in lipid metabolism, phenylalanine metabolism, amino acid metabolism, and photosynthesis were markedly enriched in YB-red vs. JC-red. We noticed that most of the DEGs mainly participated in the DNA replication and repair in the green mature stage, suggesting active vegetative growth in this period (Figure 5D). Likewise, pairwise comparisons of YB-green vs. YB-red and JC-green vs. JC-red illustrated DEGs engaged biochemical function during ripening development from green to red mature stages (Appendix A), suggesting that active alkaloid biosynthesis and flavonoid biosynthesis was closely associated with the biosynthesis of flavor substance. Convergently, we found that most of the DEGs mapped metabolic pathways were closely associated with the biosynthesis of flavor substance, implying that more vigorous metabolic events facilitated the high production of flavor substance in the JC pepper.

### 2.6. Up-Regulated Genes Related to Flavonoid and Phenylpropanoid Metabolism Promoted the High Accumulation of Flavor Substance in JC Pepper

By means of KEGG enrichment analysis, we located the capsaicin biosynthesis map and found several metabolite substrates including phenylalanine, cinnamate, 4-coumaroyl-CoA, caffeoyl-CoA, 4-hydroxy-3-methoxyphenyl-β-hydroxypropanoyl-CoA, vanillin, Malonyl-CoA, Isobutyryl-CoA, 8-methyl-6-nonenoic acid, 8-methylnon-6-enoyl-CoA and vanillylamine, which contributed to the biosynthesis of capsaicin (Figure 6A). However, the content of the above metabolites exerted no significant differences in four pairwise comparisons. Our previous omics analysis verified that metabolites and transcripts related to flavonoids and phenylpropanoids metabolism pathways were mainly responsible for capsaicin accumulation in both pepper cultivars. Gene expression involved in the biosynthesis of flavonoids and capsaicin was abstracted in an RNA-seq file and illustrated in heatmaps (Figure 6B,C). We found unique gene clusters with up-regulated transcription in both pepper cultivars. In addition, a much larger number of genes related to capsaicin biosynthesis were up-regulated in JC-red compared with YB-red, suggesting more active transcriptional events contributing to the biosynthesis of flavor substance in the JC pepper.

Subsequently, the up-regulated genes and associated metabolites were mapped into KEGG enrichment, thus showing their regulation of metabolites in flavonoids and phenylpropanoids biosynthesis. We found that phenylalanine, cinnamate, 4-coumaroyl-CoA, and caffeoyl-CoA were shared in both phenylpropanoids/flavonoids and capsaicin biosynthesis pathways (Figure 6D,E). The results suggested that the metabolic direction of these shared metabolites contributed to the accumulation of flavonoids and capsaicin in pepper. In addition, we also noticed that some structural genes regulated the biosynthesis of flavor substances, such as kaempferol, quercetin, dihydromyricetin, sinapic acid, delphinidin, and pelargonidin, were significantly up-regulated in the comparison of JC-red vs. YB-red (Figure 6D,E). The results indicated that active flavonoid and phenylpropanoid metabolic pathways also facilitated the accumulation of other flavor-related metabolites in the JC pepper. Finally, we identified four structural genes, *4CL7* (LOC107847568), *COMT* (LOC107860279), *4CL6* (LOC107869755), and *CHS* (LOC107871256), that were highly up-regulated (*p* < 0.05) in pairwise comparisons of JC-red vs. YB-red and JC-green vs. YB-green. These candidate structural genes were responsible for regulating the biosynthesis of flavor substances in the JC pepper.

### 2.7. Complex Regulation Network Involved in Regulating the Accumulation of Flavonoids and Capsaicinoids in Both Peppers

To further try to investigate the mechanisms of flavor-associated biochemical advantages in the JC pepper, we constructed the co-expression network of *4CL7*, *COMT*, *4CL6*, and *CHS* involved in the accumulation of flavonoids and capsaicin in both pepper cultivars based on Pearson’s correlation coefficient (Figure 7). We set the soft threshold to 1 (R^2^ = 0.85) to construct a scale-free network (Figure 7A). Remarkably, the expression of these genes effectively clustered the samples according to their groups (Figure 7B), suggesting their involvement in the accumulation of flavonoids and capsaicin in the JC pepper compared to the YB pepper at a different fruit stage. Then, nine modules were identified by hierarchical clustering and dynamic branch cutting (Figure 7C). Thus, these genes from 9 modules were selected as potential markers associated with the accumulation of flavonoids and capsaicin for further analysis (Appendix A). We then mapped the associated regulatory genes into KEGG enrichment pathway analysis, showing that most of them were significantly gathered in the metabolic pathway term, suggesting major regulation of metabolic changes (Figure 7D). In addition, we found that regulatory genes related to amino acid metabolism and energy metabolism were also enriched, indicating that these regulatory genes mainly functioned in their roles involved in energy production and development.

Subsequently, their expression pattern among the four groups was depicted in a heatmap referring to RNA-seq. The result illustrated that these genes were mainly represented by transcription factors (TFs), calcium-related genes, and the protein kinase (Figure 7E). Importantly, we found that most of the genes exhibited a relatively higher expression in JC-red compared to the other groups, especially in transcription factor assembly (Log_2_(foldchage) > 1). The result indicated that active expression of transcription factors may contribute to the accumulation of flavonoids and capsaicin in the JC pepper. Based on weighted correlation network analysis and gene network visualization (Appendix A), we then focused on deciphering the potential regulation of TFs that functioned in the accumulation of flavonoids and capsaicin in both peppers. As shown in Figure 7F, we found that no highly correlated transcription factors were predicted centered on *4CL6* and *COMT*, while there existed nine potential transcription factors, including ERFs, MYB, bZIP, GATA, and WRKY, that were highly associated with the expression of *4CL7* and *CHS* with the screening threshold value of Pearson correlation > 0.8 and *p*-value < 0.05 (Appendix A).

We further authorized their actual expression in both pepper cultivars by RT-qPCR, and the results showed that the four structural genes associated with capsaicin biosynthesis were significantly up-regulated in the red mature stages of the JC pepper (Figure 7F). Meanwhile, the binding sites of potential TFs for candidate genes promoter was analyzed by PLANTCARE (http://bioinformatics.psb.ugent.be/webtools/plantcare/html/) (accessed on 12 July 2023). The results showed that MYB binding sites were shared in the promoter of four candidate genes, “W-box” and “G-box” motifs were found in *4CL7* and *CHS*, which were closely associated with WRKYs and bZIP TFs (Appendix A). Based on the above results, we examined the expression level of potential transcription factors in both pepper cultivars. The results showed that MYB (LOC107850892), WRKY25 (LOC107872867), and bZIP53 (LOC107840817) were significantly up-regulated in JC-red compared to YB-red (Figure 8). These transcription factors were expected to elucidate the regulation of flavonoid-related genes contributing to the accumulation of capsaicin and flavonoids in the JC pepper in further study.

## 3. Discussion

The fruit flavor is the favored commercial characterization that determines the attractiveness and economic benefit of fruit species. And how to improve the flavor and elevate the accumulation of health-beneficial compounds in breeding programs nowadays is emphasized. Pepper is consumed in a daily diet. It contains multiple beneficial substances for people’s health, such as carotene, ascorbic acid, capsaicin, and vitamin C [28]. Of them, capsaicinoids are unique metabolites in the Capsicum genus in Solanaceae plants, which confers the addictively pungent taste in pepper [8]. To satisfy the demands for spicy pepper, capsaicin and capsaicin associated with regulatory mechanisms are stressed in molecular breeding programs, though there was limited success. Herein, our work performed the integrated analysis of metabolome and transcriptome based on *C. baccatum* and *C. frutescens* cultivars differentiated in the accumulation of flavonoids and capsaicin to illustrate the molecular mechanism for advantages of the JC pepper with increased flavor substance. These results are expected to contribute to the promotion of the JC pepper in commercial cultivation.

Flavonoid biosynthesis is an essential branch of secondary metabolic pathways, which partook in the modulation of various disadvantaged circumstances during the development of plants, such as low temperature, ultraviolet, drought, and pathogen invasion [18,19]. Increasing evidence manifested that coloration and fruit flavor were also highly correlated with the accumulation of flavonoids, contributing to plant fitness and health benefits [12]. As for pepper, the unique capsaicin substance keeps the pungent spicy in flavor. In this study, our data showed that the JC pepper exhibited the abundant accumulation of flavonoids and capsaicin compared to the YB pepper, resulting in better flavor and spicier taste as well as effective capsaicin production for medicine. Given that the JC pepper is a new pepper variety that is finished by the hybridization breeding originated from *C. baccatum* cultivar, we tried to decipher the molecular mechanism of flavor advantages. Our metabolomics showed that the JC pepper exhibited a more active flavonoid metabolism compared to the YB pepper during the transformation of the mature stage. In parallel, the highly accumulated capsaicin and analogs were observed during the red mature stage in the JC pepper, suggesting that the red mature stage is an essential period for capsaicin accumulation. Several authors demonstrated capsaicinoids and other bioactive compounds increased with fruit ripening [29,30]. Capsaicin biosynthesis belongs to the alkaloid metabolic pathways; we noticed the convergency of metabolite substrates between flavonoids, phenylpropanoid metabolism, and alkaloid metabolism. These results manifested that active flavonoid metabolism also promoted the accumulation of capsaicin in the JC pepper.

Interestingly, we found some of the flavonoid metabolites, including quercetin, daidzin, and quercitrin, which were highly accumulated during the green mature stage in the JC pepper. These metabolites not only have roles in the regulation of stress resistance in plants but also lead to excellent medical therapy for multiple human diseases [31,32]. The result suggested that the JC pepper exerted the advantage for abundant flavonoid accumulation compared to the YB pepper in both mature stages. In the assessment of new variety promotion, the variety trait involved in resistance and economic value should be fully considered. There is evidence of introgression from *C. baccatum* into other cultivars (*C. chinense* and *C. frutescens*), carrying genes conferring various biotic and abiotic stress resistances [26], which furtherly verified Xiaomila of *C. baccatum* type owned more resistances than C. frutescens. Our data showed that the accumulation of flavonoids contributed to the excellent performance of resistance and medical value in the JC pepper, implying promising prospects in JC promotion.

Deciphering the regulatory network of secondary metabolic pathways is a hot issue in recent research. The transcriptional control of flavonoid biosynthesis by the MYB-bHLH-WDR complex has been well implicated in multiple model plants [12]. Additionally, WRKY, bZIP, NAC, and ERF transcription factors were also reported to transcriptionally regulate some structural genes involved in flavonoid and phenylpropanoid biosynthesis [33,34,35]. Especially, MYB transcription factor family has been authorized to regulate several secondary metabolic pathways, the capsaicin biosynthesis included [36]. A recent work has reported that natural variations in the MYB31 promoter increase *MYB31* expression in *C. chinense* via the binding of the placenta-specific expression of transcriptional activator WRKY9 and augmentation of *CBG* (capsaicin biosynthesis gene) expression, which promotes capsaicinoid biosynthesis. In this study, our transcriptome profiles were applied to explore the expression of flavor-related genes and potential regulatory genes in both pepper cultivars. Intriguingly, the capsaicin-associated structural genes in the capsaicin biosynthesis pathway showed no significant differential expression between the two pepper cultivars, while active genes involved in flavonoid and capsaicin biosynthesis were mainly represented in flavonoids and phenylpropanoid metabolism. We found some structural genes modulating the shared metabolite substrates in both flavonoid and capsaicin biosynthesis, which promoted the accumulation of capsaicin and other flavor metabolites and was paralleled with previous metabolomics. We thus concluded that the advantage of the JC pepper showing more abundant content of flavonoids and capsaicin is mainly attributed to the active flavonoids and phenylpropanoids metabolic pathways, which have been documented in the contribution of flavor formation in many economic plants [18]. Further, we identified four essential structural genes involved in flavonoid biosynthesis, whose increased transcription in the red mature stage of the JC pepper promoted the accumulation of flavonoids and capsaicin. Liu et al. [26] found that various pungency levels in *C. annuum var. annuum* or *C. baccatum var. pendulum* were mainly achieved through the selection of different genes in the capsaicin biosynthetic pathway during their independent domestications.

Based on our WGCNA result, nine potential transcription factors, including ERFs, MYB, bZIP, GATA, and WRKY, were highly associated with the expression of flavonoid biosynthesis-related genes. Further RT-qPCR verification suggested that three transcription factors (MYB, WRKY, and bZIP) probably regulated the expression of these structural genes resulting in the variable accumulation of capsaicin and flavonoids in both pepper cultivars. It is worth noting that the JC pepper seemed to show darker red coloration on the fruit body compared to the YB pepper. Genes *4CL7*, *4CL6*, and *CHS*, which we identified, may also be used in the regulation of anthocyanin synthesis. It has been fully understood that *4CL* and *CHS* were essential structural genes that regulated the biosynthesis of 4-courmaroyl-CoA and chalcone, which were thought to be important substrates for anthocyanin synthesis and pathogen resistance [37,38]. Despite this, the finding has not been investigated; it would be enlightening to systematically authorize the molecular function of these candidate genes in multiple physiological processes and stress responses. Taken together, these potential regulatory genes activated the expression of flavonoid biosynthesis-related genes, leading to the abundant accumulation of flavor substance or involvement of other responses in the JC pepper, which deserves to be elaborated in further study.

## 4. Materials and Methods

### 4.1. Plant Materials and Measurements

Two Xiaomila cultivars (Figure 1), designated as YB (*Capsicum frutescens* L.) and JC (*Capsicum baccatum* L.), were provided by the Horticultural Research Institute, Yunnan Academy of Agricultural Science (Kunming, China). The samples used in this study were collected in Anning City (Yunnan province, China). As JC’s developmental period is shorter, at least 30 days earlier than Yunnan Xiaomila, so YB is sown one month ahead of JC to ensure they are transplanted at the same time. The fruit of both pepper cultivars in green (65 days after transplant) and red (80 days after transplant) nature stages was harvested in liquid nitrogen immediately.

To measure the levels of capsaicin compounds, including capsaicin, dihydrocapsaicin, and nordihydrocapsaicin, all samples were extracted with 95% ethanol by ultrasonication, and then they were filtered with 0.45 μm organic phase filter membrane for further chromatographic analysis. For content analysis of flavonoid, vitamin C, and phenol, the extraction protocol referred to the aluminum nitrate colorimetric method, 2, 6-dichloro-indophenol titration and folinol colorimetry assays, which were operated in multiple reaction monitoring mode (Nanjing Convinced-test Technology Co., Ltd., Nanjing, China). Each sample has six biological replicates.

### 4.2. RNA Sequencing and Analysis

Total RNA of the pepper fruit was extracted using the Trizol reagent following the manufacturer’s instructions (Invitrogen, Shanghai, China), and each group contained three biological replicates. The quality and purity of RNA was detected using a DS-11 spectrophotometer (DeNovix, Wilmington, DE, USA). The integrity of RNA was evaluated using the RNA Nano 6000 test kit of 2100 system (Agilent Technology, Santa Clara, CA, USA). The construction of messenger RNA (mRNA) libraries for each sample was instructed by Illumina NEB Next Ultra RNA library preparation kit (New England Biolabs, Ipswich, MA, USA). Then, the obtained mRNA libraries were sequenced using Illumina HsSeq-2000 platform (Illumina, San Diego, CA, USA). For data analysis, paired reads were mapped into the pepper genome referring to online database (https://pepperquence.genomics.cn/) (accessed on 5 July 2023). The mapped reads for each gene were counted by featureCounts software 2.0.2, and fragments per kilobase of exon model per million were calculated. Differential expression analysis in pairwise compared groups was performed using DESeq2 R package, and thresholds for differential expressed genes (DEGs) were restrained at adjusted *p*-value < 0.05 and │log_2_(foldchange) ≥ 1│. Gene ontology (GO) and Kyoto clusterProfiler R package. Each term in GO and KEGG analysis was defined with a significant enrichment when the adjusted *p*-value < 0.05.

### 4.3. Metabolite Extraction and Analysis

To explore the metabolic variations causing differential accumulation of capsaicin and flavor-related metabolites in the JC and YB pepper cultivars, untargeted metabolomic analyses were conducted on six independent biological replicates. A total of 50 mg of pepper fruit samples ground into a powder with liquid nitrogen was resuspended in 75% methanol and then incubated at −20 °C overnight. The samples were centrifuged at 10,000× *g* for 15 min, and the supernatant of samples was separated by Waters ACQUITY UPLCBEH Amide (2.1 mm × 100 mm, 1.7 μm). The mobile phase A of liquid chromatography is an aqueous phase composed of 25 mmol/L ammonium acetate and 25 mmol/L ammonia water, and the mobile phase B is acetonitrile. The volume of the sample tested on the machine is 1.5 μL. The column temperature is 45 °C, and the flow rate is set to 0.3 mL/min. After the sample is separated in the positive and negative ion mode, it is subjected to mass spectrometry analysis using a spectrometry (Thermo Scientific, San Jose, CA, USA). The mass spectrometry condition is set to the ion source voltage of +3.5 kV or −3.5 kV, and the ion mass scanning range is set to 70 *m*/*z*–1050 *m*/*z*. The temperature of the ion transfer tube is 320 °C, the atomization temperature is set to 200 °C, the sheath gas is 35 Arb, and the auxiliary gas is 10 Arb.

Compound Discoverer 3.0 software was used to perform peak comparison on raw data files generated by ultraperformance liquid chromatography tandem mass spectrometry (UHPLC-MS/MS). After peak area extraction, retention time correction, and feature extraction adjustments, we identify the metabolite structure through accurate mass spectrometry matching (<−25 ppm) and use MS1 and MS2 matching searches to MZcloud and ChemSpider databases to determine the metabolite structure. For the MZcloud database, metabolites are identified by accurate mass (*m*/*z*), molecular formula, and pyrolysis spectrum (MS2). The functional annotation of metabolites was analyzed by KEGG with a significant enrichment of *p*-value < 0.05, and partial least squares discriminant analysis (PLS-DA) was performed using metaX. Differential expressed metabolites (DEMs) were restrained based on variable importance in the projection (VIP) of >1, *p*-value < 0.05, and fold change (FC) of ≥2 or ≤0.05.

### 4.4. Weight Gene Co-Expression Network Analysis (WGCNA)

The WGCNA package was employed to construct gene co-expression networks using a typical set of genes involved in the biosynthesis of flavonoids, phenylpropanoids, and capsaicin, as well as potential regulatory genes in the transcriptome profiles used in this study. The analysis was performed based on the package instructions [39]. The adjacency matrix is used to compute the connection strength between each pair of nodes. Pearson’s correlation coefficient of genes was represented as the connection strength between genes. The soft threshold power of b = 1 is used to ensure the scale-free topology. Hierarchical clustering of the weighting coefficient matrix is used to define modules. Functional modules in co-expression networks with defined genes and similar expression profiles are partitioned into the same gene module using the dynamic tree pruning package. Eigenvalue and gene saliency methods are used to identify modules that are correlated with traits. The association between module eigenstates and traits is used to calculate and identify salient clinical modules. The gene significance was described as a mediated *p*-value of each gene in the linear regression between expression and traits following previous instructions [39].

### 4.5. Real-Time Quantitative Polymerase Chain Reaction (RT-qPCR) Analysis

To test transcriptome data, qRT-PCR analysis was performed on the ABI 7500 real-time PCR system (Applied Biosystems, Carlsbad, CA, USA) to identify differentially expressed genes. The primers used for qRT-PCR were designed using primer blast, and the housekeeping gene was synthesized based on the primer sequence described recurring to previous research [40]. Each sample was amplified with target gene primers and housekeeper gene primers and subjected to three biological and three technical replications, respectively. The efficiency of primer amplification was measured using LinRegPCR v2.3to detect the relative expression level of the target gene primer amplification; the housekeeping gene, β-Actin, was used to normalize the expression of target genes.

### 4.6. Statistical Analysis

Statistical significance was determined through one-way ANOVA with a Tukey test or student test analysis using Graphpad Prism 8.0. The threshold for determining significant differences was *p*-vaule ≤ 0.05. Data are provided herein as the mean ± standard deviation.

Enrichment analysis of GO function and KEGG pathway of differentially expressed genes in transcriptome data was conducted to determine the biological function and metabolic pathway of the DEG. The Pearson correlation coefficients for DEG and metabolites were calculated using R v4.2.3. Mapping DEG and metabolites simultaneously into the KEGG database to determine their common pathways. Calculate the correlation coefficient and *p*-value between DEG and metabolites. The common pathways of genes and metabolites were constructed into correlation network maps using the correlation coefficient method. Then, we use Cytascape version 3.7.1 to visualize related network diagrams.

## 5. Conclusions

The results showed that YB (*Capsicum frutescens* L.) and JC (*Capsicum baccatum* L.) were different in the accumulation of capsaicin and flavonoids. The JC pepper induced a more abundant accumulation of metabolites associated with alkaloids, flavonoids, and capsaicinoids in the red ripening stages. Four structural genes, *4CL7*, *4CL6*, *CHS*, and *COMT*, were found to be responsible for the higher accumulation of metabolites relevant to capsaicin and flavonoids. The integration of data on WGCNA, gene expression, and promoter analysis indicated that MYB, bZIP53, and WRKY25 transcription factors were potential regulators in flavor accumulation. Our findings shed light on the molecular mechanism of the accumulation of capsaicin and flavonoids in two kinds of Xiaomila and provide a comprehensive understanding and valuable information for quality pepper breeding.

## Figures and Tables

**Figure 1 ijms-25-07761-f001:**
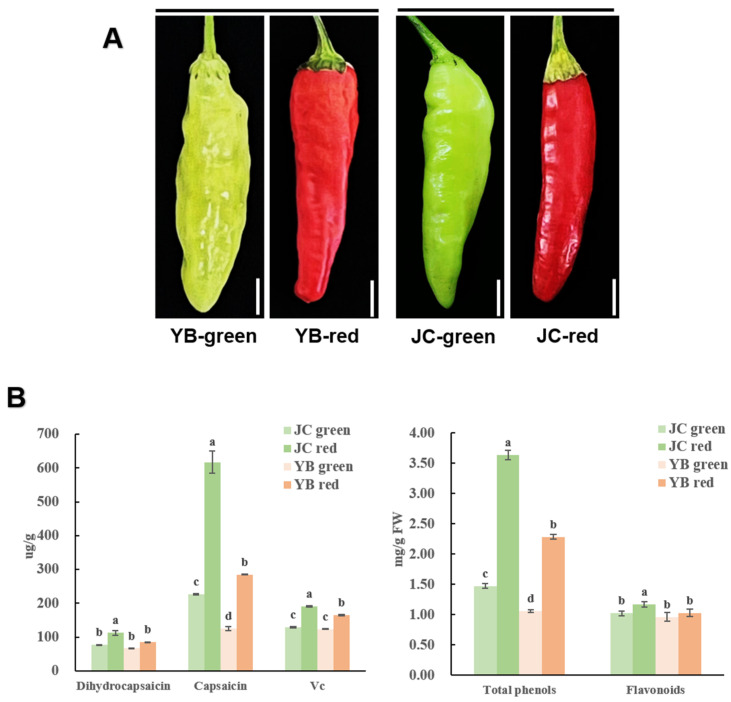
Variable accumulation of flavor-related substances was observed in JC and YB peppers: (**A**) The phenotype of both pepper cultivars in different mature stages; (**B**) The measurement of dihydrocapsaicin, capsaicin, Vc, total phenols, and flavonoids in JC and YB pepper. The different letters in the same index indicate significant differences at *p* < 0.05, according to one-way ANOVA followed by post-hoc Tukey test. Data are provided herein as the mean ± standard deviation.

**Figure 2 ijms-25-07761-f002:**
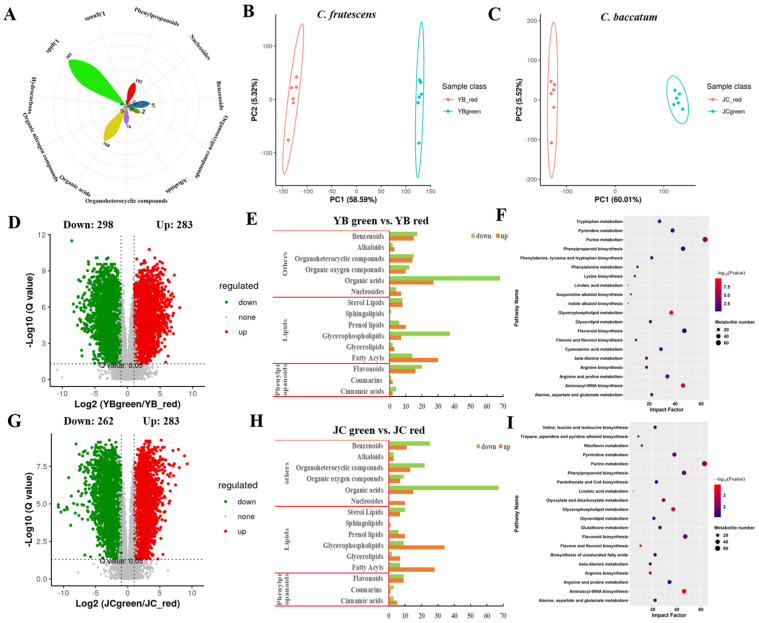
Metabolic compositions of YB and JC pepper during the green and red mature stages. (**A**) The number of different kinds of metabolites. The principal component analysis of metabolome profiles between different mature stages (green mature stage and red mature stage) in YB pepper (**B**) and JC pepper (**C**). The volcano plots depicted the expression pattern of metabolites in pairwise comparisons of YB-green vs. YB-red (**D**) and JC-green vs. JC-red (**G**). The mapping of metabolite components in the pairwise comparisons of YB-green vs. YB-red (**E**) and JC-green vs. JC-red (**H**). The KEGG enrichment analysis showed the metabolic distribution of DEMs identified in pairwise comparisons of YB-green vs. YB-red (**F**) and JC-green vs. JC-red (**I**).

**Figure 3 ijms-25-07761-f003:**
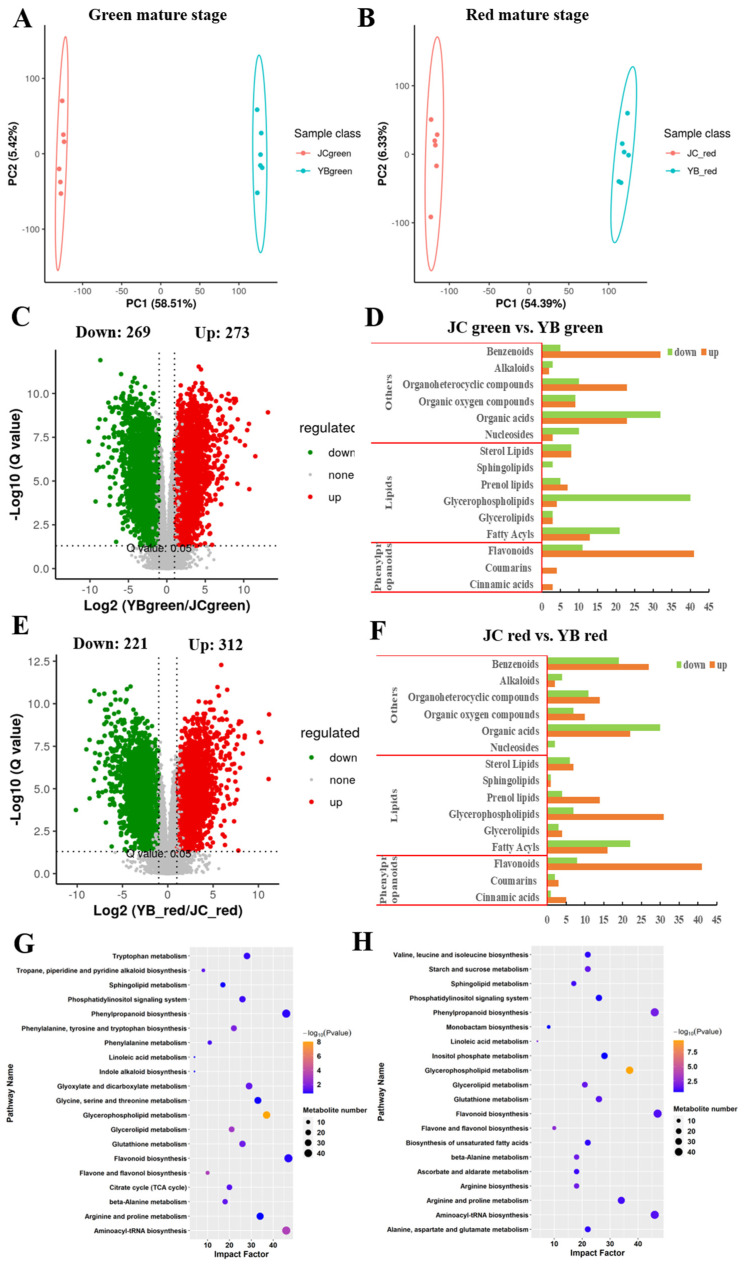
The variation of metabolome pattern in green and red mature stages between both pepper cultivars. The principal component analysis of metabolome profiles in different mature stages between YB pepper (**A**) and JC pepper (**B**). The volcano plots depicted the expression pattern of metabolites in pairwise comparisons of YB-green vs. JC-green (**C**) and YB-red vs. JC-red (**E**). The mapping of metabolite components in the pairwise comparisons of JC-green vs. YB-green (**D**) and JC-red vs. YB-red (**F**). The KEGG enrichment analysis showed the metabolic distribution of DEMs identified in pairwise comparisons of YB-green vs. JC-green (**G**) and YB-red vs. JC-red (**H**).

**Figure 4 ijms-25-07761-f004:**
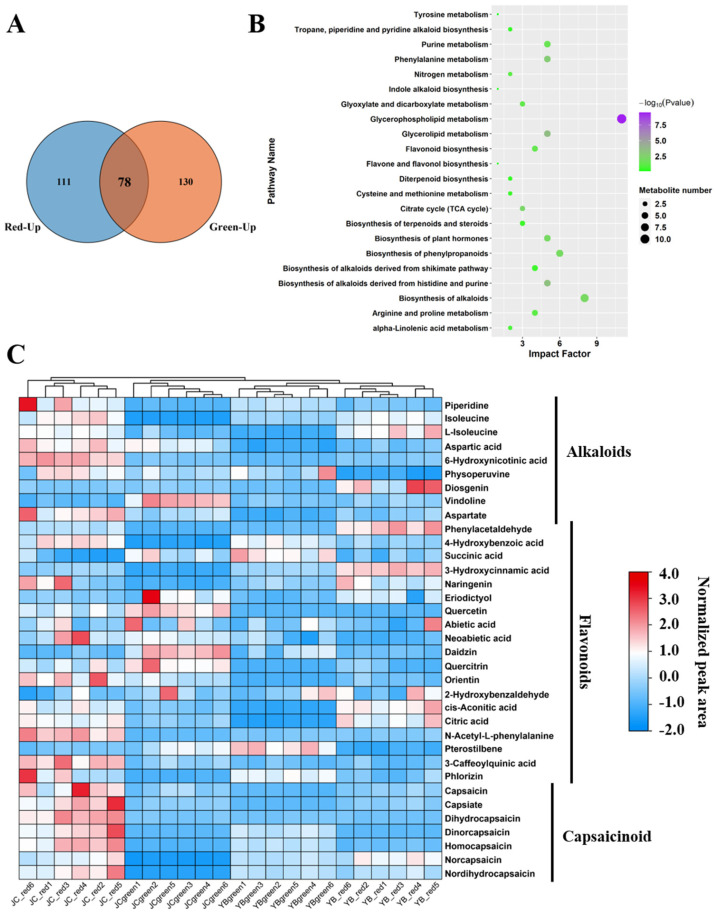
JC pepper exerted an abundant accumulation of alkaloids and flavonoids in red mature stage. (**A**) The Venn diagram showed the distribution of metabolites in the convergency of YB-green vs. JC-green and YB-red vs. JC-red. (**B**) The KEGG enrichment analysis of the shared metabolites in the comparison of YB-green vs. JC-green and YB-red vs. JC-red. (**C**) The heatmap exhibited the normalized expression of metabolites among four groups. The metabolites were divided into three groups, alkaloids, flavonoids, and capsaicin.

**Figure 5 ijms-25-07761-f005:**
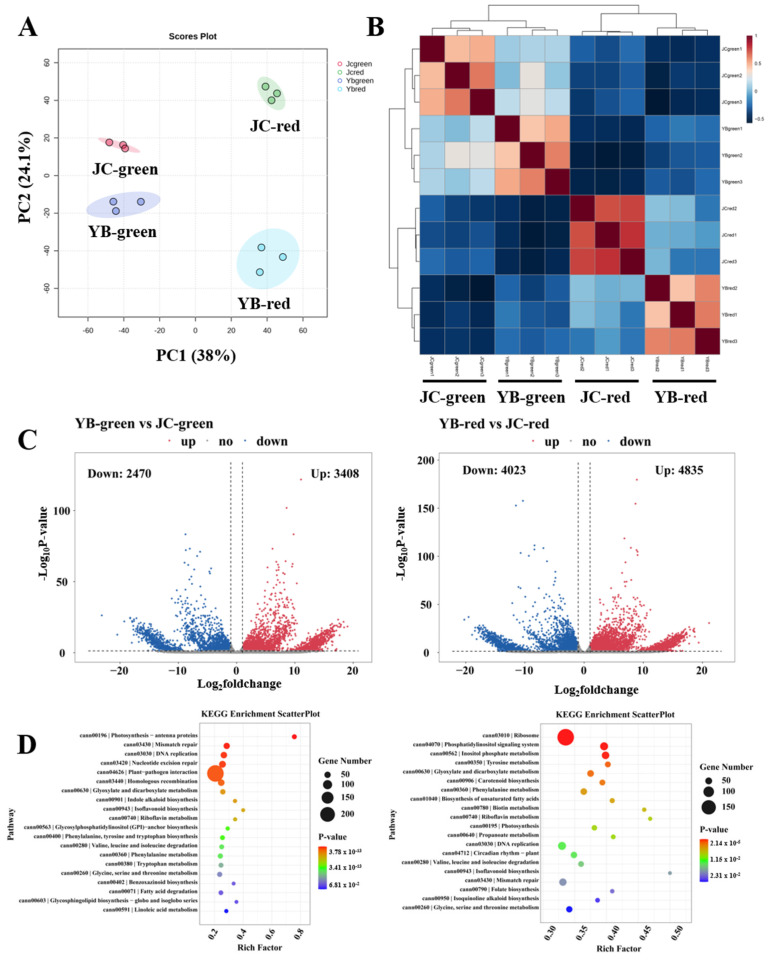
The transcriptome compositions of YB and JC peppers in different mature stages. (**A**) The principal component analysis of metabolites in different samples (YB-green, YB-red, JC-green and JC-red). (**B**) The Pearson analysis exhibited the aggregation distribution of three replicates among four groups. (**C**) The volcano plots showed the expression pattern of transcripts in pairwise comparison of YB-green vs. JC-green and YB-red vs. JC-red, respectively. (**D**) The KEGG enrichment analysis showed the metabolic distribution of DEGs identified in pairwise comparisons of YB-green vs. JC-green and YB-red vs. JC-red, respectively.

**Figure 6 ijms-25-07761-f006:**
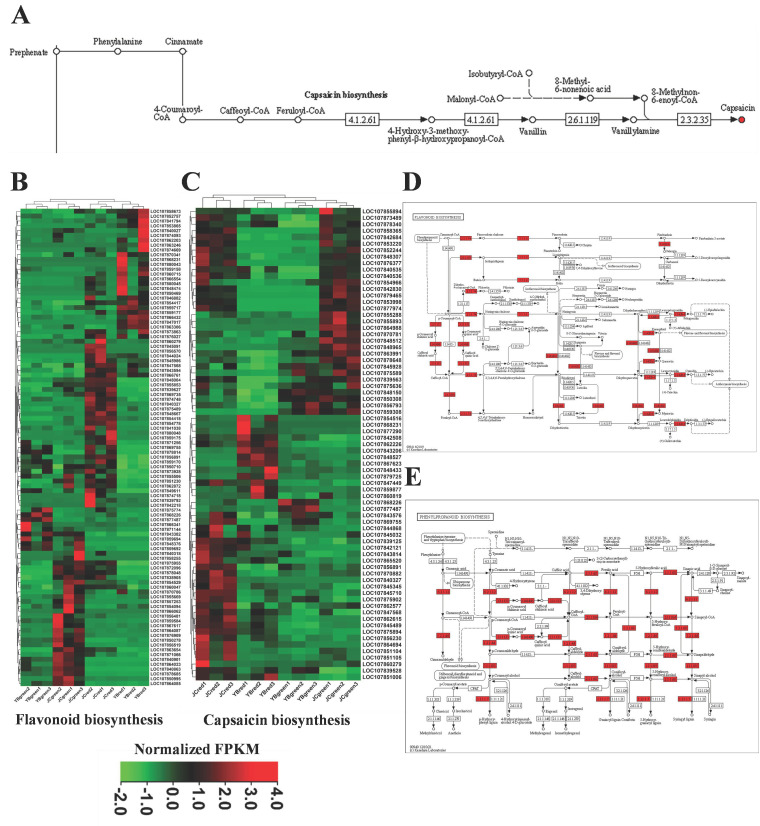
The increased transcriptional genes related to flavonoid and capsaicin biosynthesis mainly mapped in the flavonoid and phenylpropanoid biosynthesis pathways. (**A**) the depiction of biosynthetic pathway of capsaicin. The heatmaps exhibited the expression patterns of genes associated with flavonoids (**B**) and capsaicin biosynthesis (**C**) among four groups. The metabolic pathway enrichment showed up-regulated genes associated with flavonoids and capsaicin biosynthesis were mapped in flavonoids (**D**) and phenylpropanoid (**E**) metabolism pathways.

**Figure 7 ijms-25-07761-f007:**
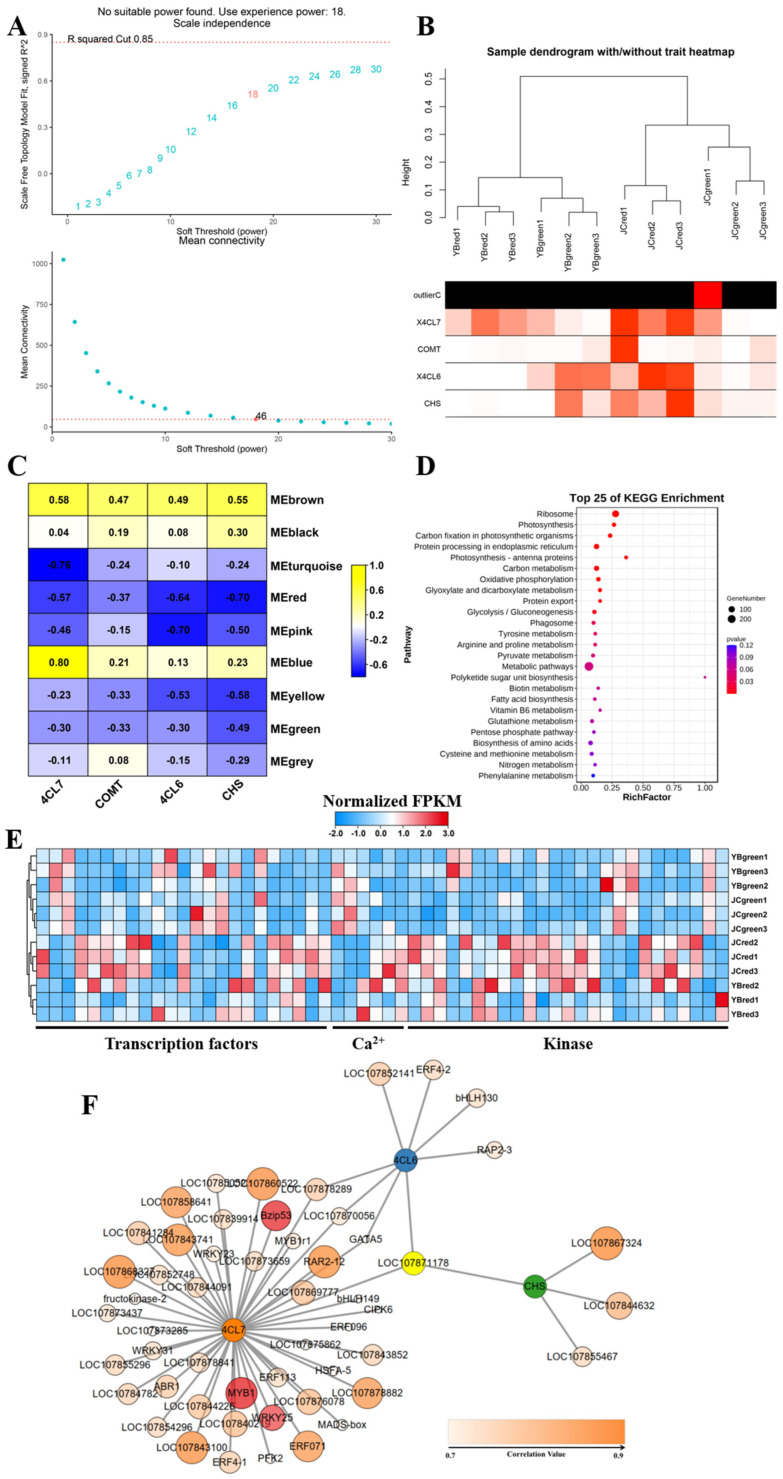
The construction of WGCNA network excavated the potential regulatory transcription factors involved in promotion for accumulation of flavonoids and capsaicin. (**A**) Network topology analysis for various soft-thresholding powers. R squared represents the goodness of fit of the expected network to the scale-free topology. When the R squared value reaches 0.85, it indicates that the network’s topological characteristics align well with the scale-free distribution. The marked numbers 18 and 46 represent the minimum soft-thresholding power needed to meet R squared = 0.85, thereby supporting subsequent specific topological network analysis. (**B**) Clustering dendrogram of samples against expression profiles of genes using WGCNA. OutlierC denotes the aberrant tissue expression samples identified during clustering analysis. The intensity of the red color blocks indicates the similarity of expression patterns in candidate genes among different samples. (**C**) Heatmap displaying the Module-trait relationships. (**D**) The KEGG enrichment analysis showed the metabolic distribution of potential regulatory genes. (**E**) The heatmap exhibited the expression pattern of potential regulatory genes among four groups. (**F**) The construction of regulatory network centered on candidate genes involved in the increased accumulation of flavonoids and capsaicin in JC pepper. The intensity of the orange color represents the expressional correlation between regulatory gene expression and candidate genes, among them, three transcription factors *BZIP53*, *MYB1* (*DIVARICATA*) and *WRKY25* were highlighted in red for satisfying the screening threshold value of Pearson correlation >0.8 and *p*-value < 0.05.

**Figure 8 ijms-25-07761-f008:**
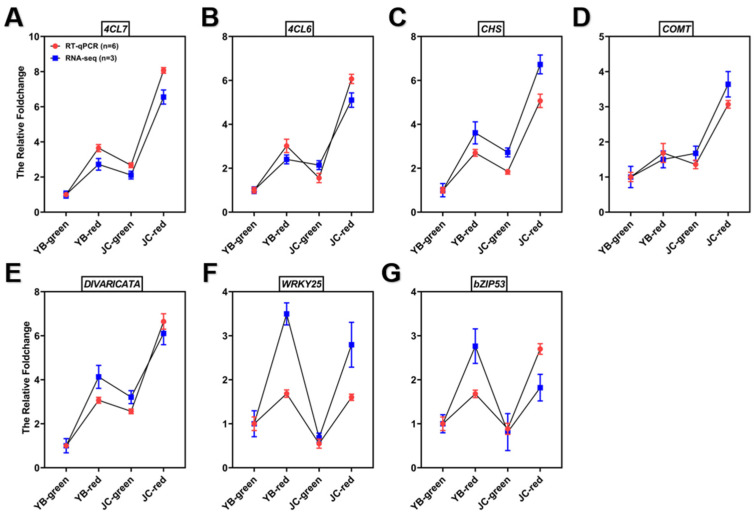
RT-qPCR verification of candidate structural genes and regulators associated with biosynthesis of capsaicin and flavonoids in both pepper cultivars. The significance was analyzed with One-way ANOVA followed by post-hoc Tukey test in above examinations; data are provided herein as the mean ± standard deviation.

## Data Availability

The data presented in this study are available in insert article or Appendix A here.

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
