# Peer review of "Dissection of Metabolome and Transcriptome—Insights into Capsaicin and Flavonoid Accumulation in Two Typical Yunnan Xiaomila Fruits"

_ijms, 2024, doi:10.3390/ijms25147761_

Round 1

Reviewer 1 Report

Comments and Suggestions for Authors

The overall research objective of the paper is clear, the plan is feasible, the design is reasonable, and the writing is standardized. But there are the following issues:

1.The line 82 indicates that Xiaomi La belongs to the CF type, but the title suggests that JC (CB type) also belongs to Xiaomi La. Please clarify and correct this.

2. The preface introduces that there are significant differences in growth and resistance between the two types of chili peppers, however the  line 466 indicates that they were both sampled at 65 and 80 days of the growth period. Is this sampling accurate? Please further clarify the sampling time.

3. Pay attention to whether the number of biological replicates in Figure 4 is consistent with the method.

4. The classification of metabolites in Figure 4 is not accurate, such as capsinate, which does not belong to the capsaicinoids.

5. No other capsaicinoids were found in Figure 1B.

6. The title mentions two typical spicy millet dishes, where are the typical ones.

7. There are many formatting issues with references, such as Latin not being italicized, etc.

Comments on the Quality of English Language

English writing needs further polishing and editing

Author Response

Dear Editors and Reviewers:

Thank you very much for your comments and professional advice. These opinions help to improve academic rigor of our article. Based on your suggestion and request, we have made corrected modifications on the revised manuscript. Meanwhile, we tried our best to improve the manuscript and made some changes to the manuscript. These changes will not influence the content and framework of the paper. And here we did not list the changes but marked in red in the revised paper. We hope that our work can be improved again. Furthermore, we would like to show the details as follows:

Suggestion from editor:

(I) Please revise your manuscript according to the referees’ comments and upload the revised file within 5 days.

(II) Please use the version of your manuscript found at the above link for your revisions. 

(III) Please check that all references are relevant to the contents of the manuscript.

(IV) Any revisions to the manuscript should be highlighted, such that any changes can be easily reviewed by editors and reviewers.

(V) Please provide a short cover letter detailing your changes for the editors’ and referees’ approval.

(VI) If the reviewer(s) recommended references, please critically analyze them to ensure that their inclusion would enhance your manuscript. If you believe these references are unnecessary, you should not include them.

Our response to the editor:We have downloaded the version of manuscript found at the above link for revisions. Revised portion are marked in red in the paper. All references have been checked and determined to be relevant to the manuscript. A short cover letter detailing our changes for the editors’ and referees’ approval was provided. The main corrections in the paper and the responds to the reviewer’s comments are as following:

Responds to the reviewer’s comments:

Reviewer #1:

1.The line 82 indicates that Xiaomi La belongs to the CF type, but the title suggests that JC (CB type) also belongs to Xiaomi La. Please clarify and correct this.

 The author’s answer: We sincerely thank you for your careful reading. Xiaomila belongs to C. frutescens L. species. However, there are issues in the industrialization process, including variety degradation, low yield, delayed growth period, and weakened resistance. Importantly, artificial pollination and pollen collection are challenging due to its small flowers. Therefore, the limited agronomic measures and breeding techniques for Xiaomila merely maintained its purification and sub-generation. To address this issue, a new variety of Xiaomila designated as JC (Jingcui), was generated through the hybridization of C. baccatum x C. baccatum. Thus, JC belongs to a new type of Xiaomila. Both types of Xiaomila have become the main varieties in Yunnan now. This information we have explained detailly at the line 96-97 with red mark. Thank you again for your attention and revision.

2.The preface introduces that there are significant differences in growth and resistance between the two types of chili peppers, however the line 466 indicates that they were both sampled at 65 and 80 days of the growth period. Is this sampling accurate? Please further clarify the sampling time.

The author’s answer: We apologize for the unclear writing. JC's growth period is at least 30 days shorter than Yunnan Xiaomila. Therefore, YB is sown one month ahead of JC to ensure they are transplanted simultaneously. This ensures consistency in the growth stage of peppers during sampling. To avoid misunderstanding, we have provided further clarification in the revised manuscript on lines 480-484. Thank you again for your careful check.

  1. Pay attention to whether the number of biological replicates in Figure 4 is consistent with the method.

The author’s answer: We are very sorry for our negligence. We have made further confirmation that each group has six biological replicates, consistent with our previous analysis in Figure 1-Figure 4. Untargeted metabolomic analysis were conducted on six independent biological replicates. Three biological replicates of each group were used for RNA-seq. We have amended our error in paper line 492, and checked it in full text. Thank you again for your careful revision.

  1. The classification of metabolites in Figure 4 is not accurate, such as capsinate, which does not belong to the capsaicinoids.

The author’s answer: Thank you for your careful revision. Indeed, capsinate belongs to Capsinoid, which is different from capsaicinoids. However, we hold a differing view on this matter. While as a capsaicinoid-like compound, capsinate shares a similar biological structure with capsaicinoids but showing low pungency (depicted in figure below). In vivo, both shared biosynthetic pathways including phenylalanine metabolism, vanillylamine metabolism and so on. Studies have shown that the capsaicin synthase gene Pun1 controls the synthesis of both capsaicinoids and capsiate compounds (Setwart et al. 2005). Some researchers also suggest that capsinate compounds can be broadly classified as capsaicinoids. Thus, in our initial metabolome analysis, the identification of capsaicinoid metabolites contained both capsaicin and capsiate compounds. We hope you can understand our perspective. Here, to avoid any ambiguity, we have redefined capsinate description, and related additions were detailed in line 227-278. We redrew Figure 4C image and corrected “capsaicin” to “capsaicinoid” substance. Thank you for your rigorous revision again.

the structure of capsaicin and capsiate

Ref: Stewart, C., Jr, Kang, B. C., Liu, K., Mazourek, M., Moore, S. L., Yoo, E. Y., Kim, B. D., Paran, I., & Jahn, M. M. (2005). The Pun1 gene for pungency in pepper encodes a putative acyltransferase. The Plant journal : for cell and molecular biology, 42(5), 675–688. https://doi.org/10.1111/j.1365-313X.2005.02410.x

  1. No other capsaicinoids were found in Figure 1B.

The author’s answer: Thank you for your careful revision. In this study, we selected capsaicin and dihydrocapsaicin as the primary indicators of pepper capsaicinoids because they account for more than 80% of the total capsaicinoid content in Capsicum fruit. Therefore, we considered that both can be used as standards for assessing the degree of pungency in peppers. We hope you will appreciate our point of view. Thank you for your revision again.

  1. The title mentions two typical spicy millet dishes, where are the typical ones.

The author’s answer: Thank you for your careful revision. We are very sorry for our unclear description. In our study, YB (Capsicum frutescens L.) and JC (Capsicum baccatum L.) are the two important indigenous varieties of Xiaomila in Yunnan, which were widely planted in most of pepper production regions. Interestingly, the varying levels of pungency in the two types of peppers attract different consumer groups. We observed that JC peppers have a more attractive pungency. Integrated analysis of metabolome and transcriptome was used to decipher the potential molecular mechanism of spicy flavor between them. We also attempted to use terms like "local" or "indigenous" to emphasize the significance of both pepper cultivars, but "typical" seems more accurate. Consequently, we have named our paper as “Dissection of Metabolome and Transcriptome Insights into the Capsaicin and Flavonoids Accumulation in Two Typical Yunnan Xiaomila Fruits”. We hope the you can understand our perspective. We would appreciate a more precise term to enhance the accuracy of our description. We look forward to your feedback. Thank you for your careful revision again.

  1. There are many formatting issues with references, such as Latin not being italicized, etc.

The author’s answer: We are very sorry for our careless mistakes. Thank you for pointing this out. We have checked them all and made modifications. And corrected reference formats were marked with red. Italicize the latin letters in reference 5, 11, 37 and 40.

Thank you again for your positive comments and valuable suggestions to improve the quality of our manuscript.

Reviewer 2 Report

Comments and Suggestions for Authors

The manuscript analyzes the capsaicin and flavonoids accumulation in two pepper cultivars belonging to Capsicum frutescens L. and Capsicum baccatum L. Pepper is an important economic horticultural species. The work aimed to compare the difference of flavor accumulation between two pepper genotypes at different development stage by multi-omics analysis. The authors findings provide a comprehensive information for pepper potential breeding programs. The manuscript is generally well structured. The results are interpreted statistically.

 However, there are some shortcomings:

-     explain in the legend the meaning of the bars in the graphs and the letters in the data tables

-        include more details on the cultivars used in the study

-        the text from lines 564 to 578 was inserted by mistake???

-     more details must be presented regarding the software used for data processing.

Author Response

  1. Explain in the legend the meaning of the bars in the graphs and the letters in the data tables

The author’s answer: We sincerely thank you for your careful reading. As suggested, we added an explanation of the meaning of the bars in the graphs and the letters in the data tables in line 126-128 and line 381-384. Thank you again for your careful revision.

  1. Include more details on the cultivars used in the study

The author’s answer: According to your nice suggestions, we have made extensive corrections to our previous draft, the detailed description about two cultivars was added in line 87-99. Thank you for your valuable comment again.

  1. The text from lines 564 to 578 was inserted by mistake???

The author’s answer: We are very sorry for our negligence of this part. We deleted this part and added more detailed information about data processing. Thank you for your careful check again.

  1. More details must be presented regarding the software used for data processing.

The author’s answer: We greatly appreciate your valuable suggestions to improve our paper. We have included details about the software used for transcriptome data processing, fluorescence quantitative data analysis, and co-expression network construction in our study. Adding this information as per your request enhances the reference value and significance of our research paper. Thank you again for your valuable suggestion.

Thank you very much for your attention on time. Look forward to hearing from you.
